# Evaluation of Picker Discomfort and Its Impact on Maintaining Strawberry Picking Quality

**Piotr Komarnicki [1] and Łukasz Kuta [2],***

1   Institute of Agricultural Engineering, The Faculty of Life Sciences and Technology, Wrocław University of Environmental and Life Sciences, 51-630 Wrocław, Poland; piotr.komarnicki@upwr.edu.pl
2   Institute of Environmental Protection and Development, The Faculty of Environmental Engineering and Geodesy, Wrocław University of Environmental and Life Sciences, 50-363 Wrocław, Poland
*   Correspondence: lukasz.kuta@upwr.edu.pl; Tel.: +48-71-320-55-85

**Abstract:** In this paper, the authors present the relationship between the assumptions of ergonomics in the work of a strawberry picker and quality of picked fruit. The body posture that a person adopts while working has a significant impact on their health, working comfort, and productivity, but also on the quality of the fruit that is harvested. This paper identifies three characteristic picker positions during strawberry harvesting. A synchronized surface electromyography (sEMG) instrument together with the Tekscan® surface pressure measurement system allowed for the determination of the influence of working position on changes in the load of the picker's musculoskeletal system and the surface pressure exerted on the fruit during manual strawberry picking, which are decisive factors for maintaining fruit quality. In addition, compression tests on whole strawberry fruit were carried out as a benchmark to evaluate and compare the maximum forces as well as the destructive pressures on the fruit. From the tests, we found that the most comfortable position of the worker's body was determined along with the harvesting technique (position during work) that has the least negative effect on the quality of the harvested fruit. Consequently, the level of dynamic load on the worker was determined.

**Keywords:** manual harvesting; surface pressure; discomfort at work; sEMG; muscle; strawberry

## 1. Introduction

The strawberry is a fruit with no blossom fall and with a limited harvesting period. Due to its fragile structure and physical properties, it is highly susceptible to mechanical damage and has a low post-harvest life [1]. Aliasgarian showed that the picking operation accounts for about 51% of the damage to strawberry fruit throughout the production stage [2]. Mechanical handling in the supply chain results in increased fruit spoilage and thus handling requirements to avoid quality loss [3–6]. Consequently, knowledge of fruit biomechanics forms an important basis for the post-harvest assessment of strawberry quality, allowing the prediction of the intrinsic mechanical response (damage evolution) [7–9]. The study of the behavior of biological materials under dynamic and impact loads generated by picking is not widely reported in the literature, especially with regard to contact issues. For apples, it has been shown that the surface pressure generated during manual harvesting has an important influence on fruit quality [10,11]. Information on the distribution of surface pressures allows a quick assessment of the behavior of biological material under quasistatic as well as dynamic loads. Marshall, by measuring ultrasonic waves, obtained surface pressure distributions as a function of the loading force and pressure distributions along the contact surface [12]. Maximum surface pressures were recorded at the central point of contact between the apple and the loading device. Herold, on the other hand, showed that this happens only up to a certain limit value, above which there is a sharp decrease in the load bearing capacity of the tissues located at the central point of

contact, and the maximum pressure values are at the edge of the point of contact [13]. This limit is considered to correspond to the local damage of the bioyield point [14,15].

Undoubtedly, the effectiveness of a person's work is determined by the conditions of the working environment. In many cases, working in uncomfortable conditions causes permanent damage to health. In order to take effective preventive action against musculoskeletal overloading, such risks must be assessed and identified as early as when the work station is designed [16]. The load assessment can be conducted with a number of available methods. For example, electromyography, a method commonly used in medicine, can be used to assess dynamic loads in agriculture. On this basis, attempts have already been made to design workstations in Europe, among others [17–19]. Other methods of load assessment include the pulmonary ventilation method, heart rate index, and the number of heart beats per minute before and during exercise. Thetkathuek concluded that work-related musculoskeletal disorders are cumulative disorders that are common in farmers [20]. They noted that pain regularly occurs in the cervical section of the spine and in women in the lower back. Based on these studies, they recommended the determination of musculoskeletal diagnoses in more detail. In turn, Mokhtar described that the risk of developing work-related musculoskeletal disorders (MSDs) in agricultural activities is very high [21]. This is a local and global problem that is currently not well recognized. In the studies, they used RULA analysis. From the results, it was found that 17% of the harvesters scored 5 points in their work assessment, which means that a change in body posture must be carried out immediately. Moreover, they noted that repetitive work is also the main factor contributing to musculoskeletal disorders. The study by Kim aimed at determining the prevalence of upper limb musculoskeletal disorders (MSDs) and identifying disability factors among fruit tree growers in Korea [22]. The total number of examined growers was 460. The survey indicated that nearly 60.4% of growers complained about MSD disorders. They proposed that not only should farmers be educated but more effective activities should be implemented. In turn, Kuta conducted a study where the main objective was to investigate the workload during manual and mechanized agricultural tasks [23]. The analysis was conducted on 15 farmers on their own farms, with regards to morning and evening milking, in the tethering and carousel systems and included lifting and carrying a full bucket or a sack and driving a tractor. The analysis of muscle load was conducted with the surface EMG (electromyography) system and Job Strain Index method. Młotek noted that the generally preferred approach is manual fruit picking assisted by modern technology associated with the movement of the picker with the use of a mobile platform and mobile lifting baskets [10]. Due to many hours of long physical strain as well as the repeated nature of the picker's movements, apple harvesting may contribute to the development of ailments in the musculoskeletal system. Ng evaluated the association between the self-reported prevalence of musculoskeletal disorders and productivity. They also studied the impact of musculoskeletal disorders on the productivity of workers. Moreover, they examined four different aspects: daily harvesting quantity, efficiency score, sick leave, and productivity at work. Ng further noted that musculoskeletal disorders pose a global problem [24].

There is a lack of research results in the literature that describe the effect of the ergonomics of picking sensitive strawberries on the surface pressure exerted by the picker. Different manual techniques for picking, transporting, and sorting fruit help to determine the optimum body position for workers during harvesting to minimize musculoskeletal strain. Therefore, the aim of this research is to determine the influence of working position on changes in the load of the picker's muscles and the surface pressure generated during hand contact with the fruit, which are decisive in maintaining the quality of biological material collected.

## 2. Materials and Methods

### 2.1. Field Test—Evaluation of Pressures

The tests were conducted in two stages. In the first stage, field tests were carried out, during which strawberries of the Clery variety were manually harvested from a plantation located in Poland, near the city of Wrocław. This variety of strawberry was used for research purposes because of its increasing popularity in Poland. The fruit was grown in open fields in raised beds lined with straw. It was harvested in June 2021 at full red harvest maturity at a temperature of $28 \pm 1$ °C and relative humidity of 50%. The field experiments consisted of the simultaneous measurement of surface pressures and recorded picker muscle tensions (sEMG). The surface pressure exerted on the harvested fruit was tested using a portable Tekscan® system (South Boston, MA, USA) [13]. The device consisted of an ultra-thin pressure sensor (model 5027) attached directly to the picker's index finger (Figure 1).

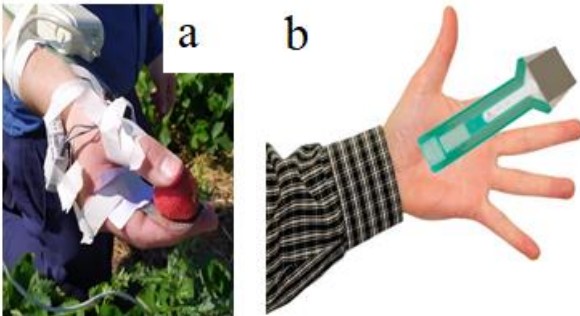

**Figure 1.** Picker's hand equipped with the sEMG (**a**) and the Tekscan (**b**) surface pressure measurement system.

The pressure sensor which was used had a pressure range of 0.345 MPa, a working area of $27.9 \times 27.9$ mm, a thickness of 0.1 mm, and a sensel density of 248 sensels cm$^{-2}$. Data transmission to the computer took place via a multi-channel hub (VersaTek 8 Port Hub) connected directly to a special handle (VersaTek Sensor Handles), inside which a pressure sensor was placed. The system, together with the I-Scan software, enabled real-time data recording at sampling rates of up to 5 kHz. The field measurements were carried out in five repetitions for each of the three harvesting positions (1—squatting, 2—kneeling, and 3—upright). A single measurement lasted 2 min, during which 15 to 20 fruits were collected. In total, around 300 strawberry fruits were collected for testing. The contour images recorded by the Tekscan® system were considered for the phases of occurrence of maximum surface pressures. The forces and contact surfaces thus obtained depended on the maximum surface pressures resulting from the characteristic structure (the presence of achenes over the entire fruit surface) (Figure 2a,b).

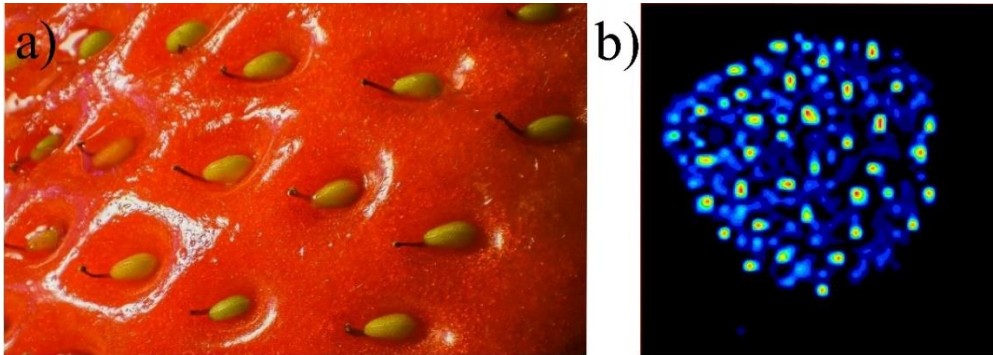

**Figure 2.** View of the achenes on the fruit surface (**a**) and their resulting pressure concentration (**b**).

The measurement errors of the Tekscan® system were mainly due to the geometry of the strawberry fruit, the contact force, and the contact area (the change in contact area of the radius of curvature did not exceed 2%). Due to the precise selection of the study samples, overall variation was ignored.

### 2.2. Surface Electromyography

One of the research methods used for the purpose of this article was surface electromyography. This is an innovative method for determining muscle tension values (mV) during exercise [25]. The device has international SENIAM and ISEK certificates confirming the accuracy of the measurements [26]. It consists of software on a portable computer, four wireless sensors (four-channel device), and gel electrodes that are non-invasively attached to the skin of the test subject. Prior to measurement, the skin of the test subject is thoroughly cleansed to remove potential contaminants. On the basis of the sEMG, the forces generated by muscle activity during successive phases of the actions can be determined. Electromyography can be used to assess the tension of all human muscle groups. Measurement results are given in millivolts (mV). The measurement error of the device is 2 mV. The mean range of muscle static potential is –40 mV to +40 mV during muscle engagement with light work. Surface electromyography, in addition to the analysis of spatio-temporal parameters and kinematic quantities, is used to determine the correct posture during physical activity. While measuring selected parameters, the program records and saves the generated results in real time. Figure 3 shows an example of the result window obtained with the EMG software (NORAXON, Phoenix, AZ, USA). Due to the fact that the device has the ability to test four muscle groups simultaneously, an example of the muscle tension (μV) results obtained as a function of time is presented. The graph clearly shows the increase in muscle tension when the load on the test subjects increases during work. The system allows the range of the result scale to be specified, the scope of the study to be selected, and the markers to be analyzed to be defined.

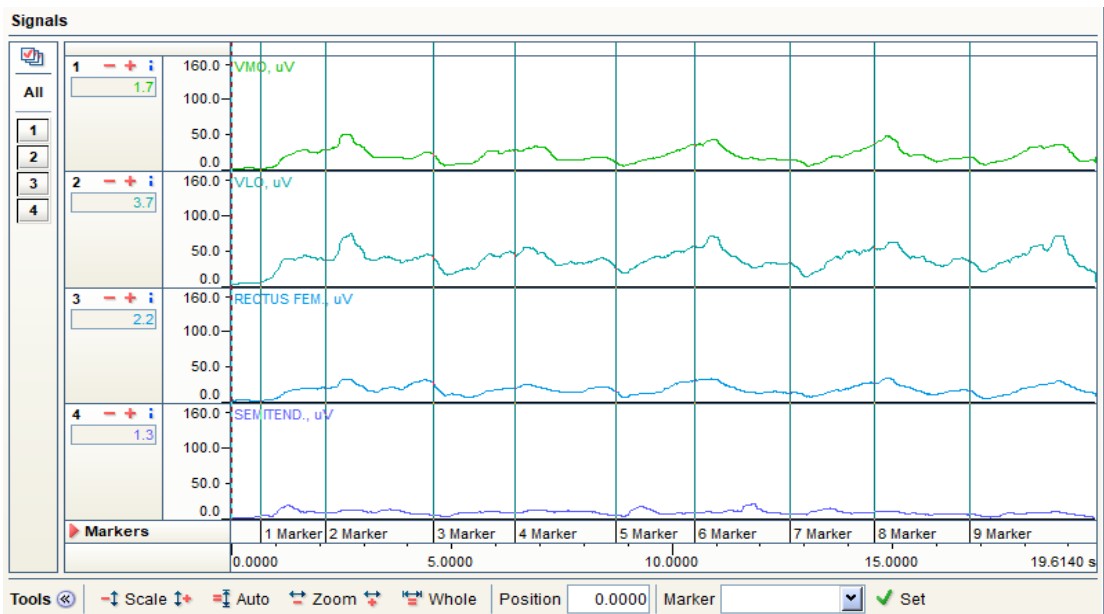

**Figure 3.** Results window generated by the sEMG system when testing human muscle load over time.

Four muscle pairs were designated to measure the musculoskeletal load of the strawberry picker. These muscles were identified as the motor units that drive a specific segment of the picker's body during fruit picking. An additional reason for this choice was the negative sensations and pain centers arising in these areas in connection with the manual work carried out. The muscle groups identified (Figure 4) are as follows:

1.    Muscles of the lumbar spine—right and left;
2.    Abductor pollicis;
3.    Metacarpal muscle.

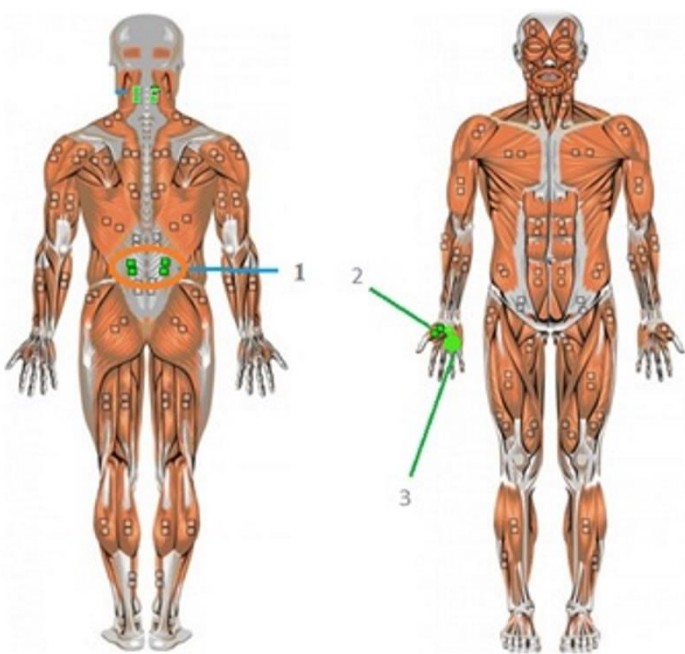

**Figure 4.** Strawberry pickers' muscle groups tested: 1—Muscles of the lumbar spine; **right** and **left**. 2—Abductor pollicis brevis. 3—Metacarpal muscle.

The article evaluates the load of selected segments of the musculoskeletal system of the strawberry picker in three harvesting positions which they usually adopt during work. The body posture of the picker is shown in Figure 5a–c below. In addition, the characteristic position of the individual musculoskeletal segments during work is shown by means of lines.

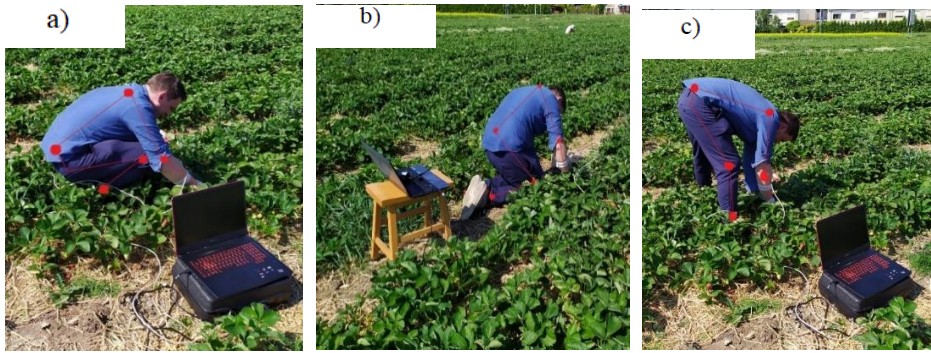

**Figure 5.** Tested body positions of the strawberry picker during strawberry picking: (**a**)—squatting position, (**b**)—kneeling position, (**c**)—upright position.

### 2.3. Laboratory Tests—Compression Tests

In the second stage of the research, the fruit was transported to the agrophysics laboratory of the Institute of Agricultural Engineering, where on the same day, the material was selected in terms of geometric and mass aspects, and the firmness of 45 selected strawberry fruits was examined. The laboratory temperature was $25 \pm 1$ °C and the relative humidity was 50%. The weight of a single fruit was determined using electronic scales (AXIS, AD500, Wrocław, Poland) with a range of 500 g and accuracy of 0.001 g. Regarding geometric measurements, the height and mean diameter of the fruit were determined using

electronic calipers with an accuracy of 0.01 mm (Hogetex, Varsseveld, The Nederlands). On the basis of the linear dimensions of the strawberries, the shape sphericity coefficient was determined according to Formula (1), as proposed by Mohsenin [27].

$$\varphi = \frac{\left(LD^2\right)^{0.333}}{L} \tag{1}$$

where

$D$—diameter of strawberry (mm);
$L$—length of strawberry (mm);
$\varphi$—sphericity of strawberry (–).

The harvest ripeness of the flesh was determined in firmness tests using a digital fruit firmness penetrometer (GY-4, by Newtry, Huizhou, Guangdong, China), with an accuracy of 0.01 N, with a 3.5 mm stem diameter designed for soft fruit. The penetrometer was mounted in a lever handle, ensuring repeatable displacement conditions at a similar speed and force. Whole fruit compression tests were conducted on a selected group of strawberries using an Instron 5566 testing machine (Norwood, MA, USA) integrated with the Tekscan® surface pressure system (Figure 6).

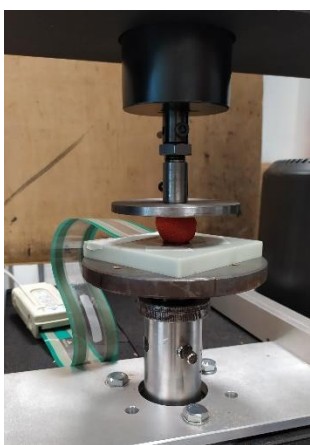

**Figure 6.** Compression test integrated into the Tekscan® surface pressure measurement system.

The fruit was placed in a lateral position and subjected to compression until failure on a non-deformable substrate for three head loading rates (1, 10 and 100 mm·min$^{-1}$). The above tests allowed the simultaneous measurement of failure loads, deformations, contact surfaces, and maximum surface pressures. The tests were carried out in 15 repetitions with 45 fruits.

### 2.4. Statistical Analysis

The data obtained were statistically processed in Microsoft Excel. This software allows the determination of the mean values, deviations, and standard errors of the data obtained. Additionally, a Student's t-test was performed in the STATISTICA 12 software (StatSoft Polska Sp. z o.o., Kraków, Poland) to analyze the significance of differences in loads depending on the body position of the picker. For statistical purposes, differences were assumed to be statistically significant when $p < 0.05$ (*p*-value probability). In addition, correlation coefficients were determined between the load values of individual muscles of the strawberry picker and the value of contact forces $F$, contact area $A_c$, and the value of pressures $p_{max}$ for the studied body positions of the picker.

## 3. Results

*3.1. Field Test Results—Surface Electromyography*

Figures 7–9 show the results of loading the different muscle groups of the strawberry picker during 60 s of work. The results are presented as a percentage of maximal muscle strength (% MVC).

### 3.1.1. Squatting Position

Figure 7 below shows the load values of the muscles tested when picking strawberries in a squatting position. During moments of contact between the picker's hand and the fruit, increments in the load on the abductor pollicis muscle are noticeable to a level not exceeding 18% of the MVC. Due to the fact that the thumb is treated as a stop finger, the load values increase when the fruit is detached from the stem. In the case of the lumbar muscles, there is a noticeable increase in load when the body leans forward while picking the fruit. The maximum values do not exceed 20% of the MVC. The lowest values were observed for the metacarpal muscles located on the middle part of the hand. The values in this case are in the range of 2–6% of the MVC.

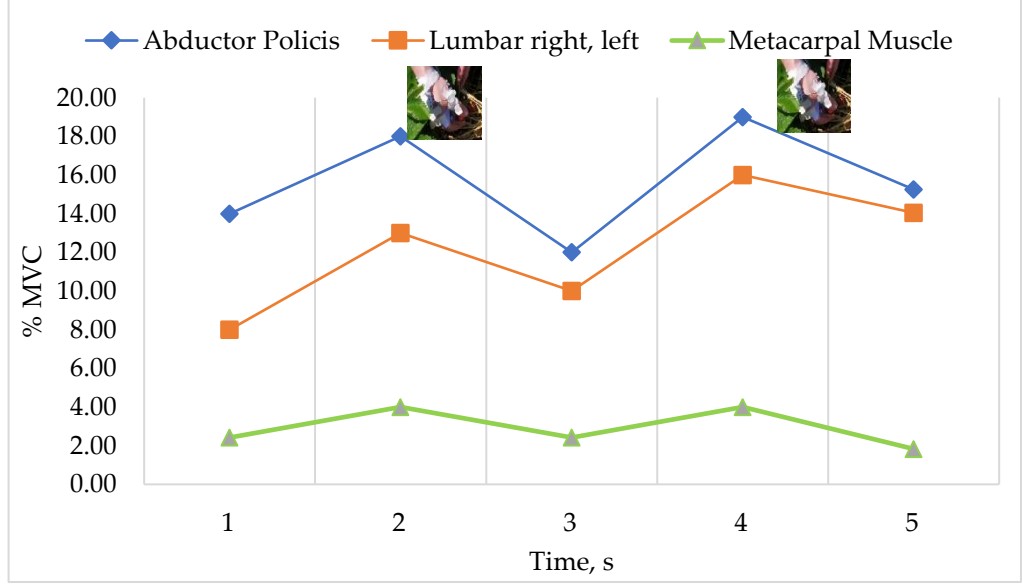

**Figure 7.** Load on the test muscles of a strawberry picker in a squatting position.

### 3.1.2. Kneeling Position

Figure 8 shows the muscle load of the strawberry picker in the kneeling position. In this case, the lowest load was observed for the metacarpal muscle (4–6% MVC). The load on the lumbar muscles varied cyclically depending on the position of the back. When leaning forward, the load increases by about 8–10% of the MVC. It can be seen from the graph that the mean load level increases over the period studied, indicating fatigue in the lumbar muscles when holding this position during harvesting. The load on the abductor pollicis muscles varies during the working cycle. The load level increases when the picker's hand comes into contact with the fruit, and the maximum cyclic loads do not exceed 30% of the MVC.

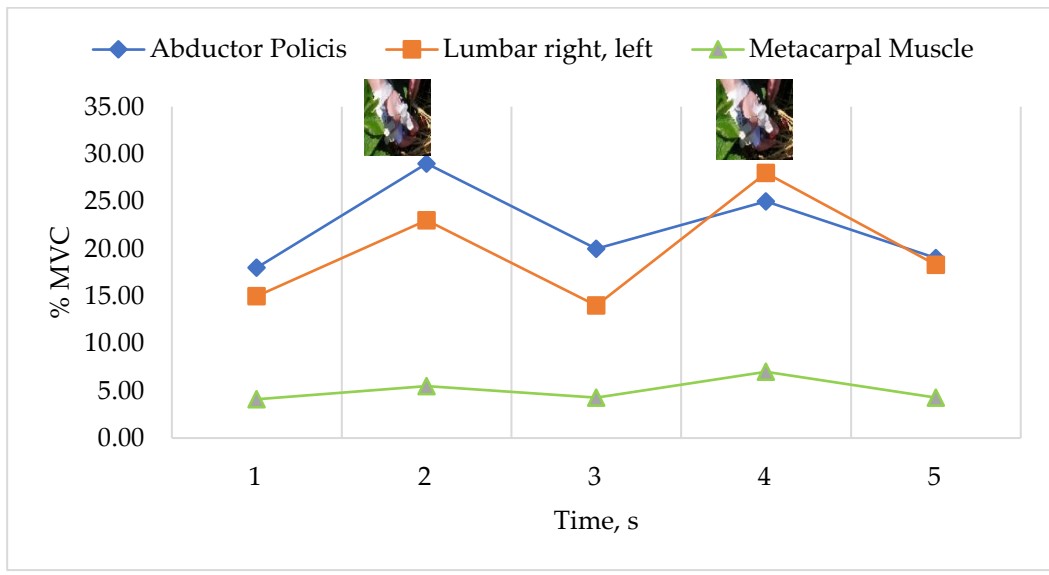

**Figure 8.** Load on the test muscles of a strawberry picker in a kneeling position.

### 3.1.3. Position with Straight Legs

On average, the highest level of load on the muscles tested occurred in the position when the picker had their legs straight. This is a very uncomfortable body position that forces a deep forward bend, which is evidenced by the values of load on the lumbar muscles, reaching values of 45% of the MVC. Figure 9 shows that during harvesting, the load on this muscle group increases proportionally as a consequence of their increasing fatigue. The load on the abductor muscles increases cyclically while gripping the fruit. Here, there is clearly a forced range of motion of the arms, which further increases the level of strain. On contact with the fruit, the load value rises to 38% of the MVC. The lowest level of loading is shown for the metacarpal muscles. In this case, the load exceeds 10% of the MVC and continues to gradually increase.

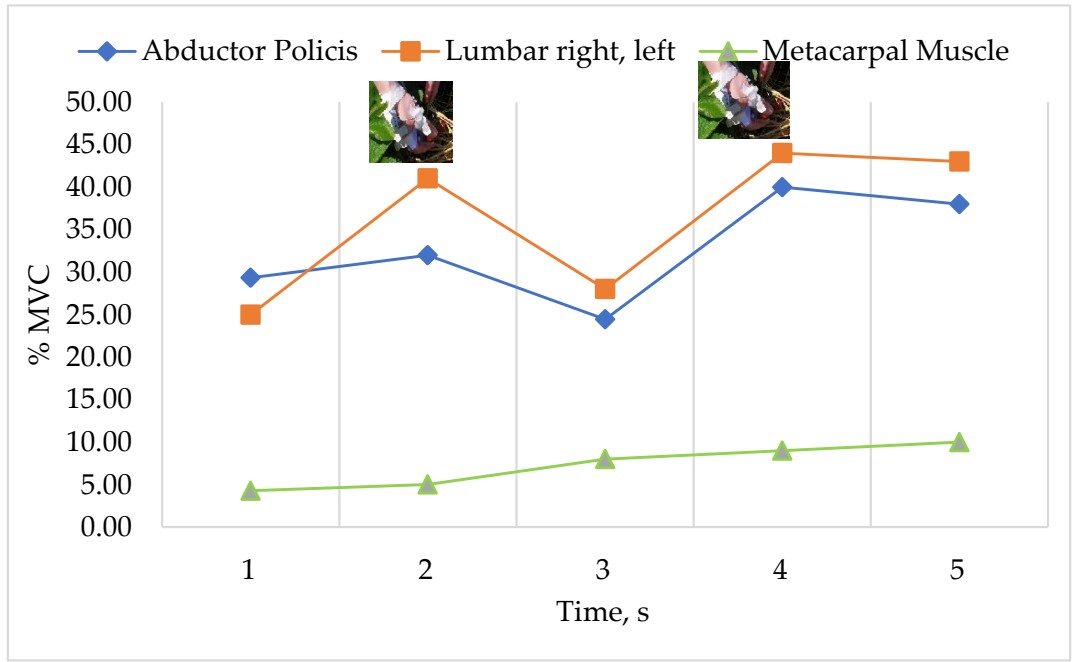

**Figure 9.** Load of the test muscles of the strawberry picker in a straight-legged position.

Table 1 presents a summary of the load on the individual muscles of the strawberry picker depending on the body position taken. According to the load waveforms in Figures 7–9, the lowest mean loads on the examined segments of the musculoskeletal system were observed in the squatting position, while the highest loads were observed in the straight-legged position.

**Table 1.** Mean values and standard deviations of the load on the tested muscles of the strawberry picker.

| Body Position | | Load Measure % MVC, Muscle Tested | | |
|---|---|---|---|---|
| | | Abductor Pollicis | Lumbar Right, Left | Metacarpal Muscle |
| Squatting | Mean | 13.79 | 13.48 | 3.04 |
| | SD | 1.49 | 1.53 | 0.71 |
| Kneeling | Mean | 23.30 | 22.99 | 4.19 |
| | SD | 2.57 | 3.78 | 0.72 |
| Straight legs | Mean | 29.73 | 32.96 | 4.16 |
| | SD | 4.79 | 5.81 | 0.47 |

Based on the t-Student test, the lumbar and abductor pollicis muscle loading scores were found to be statistically significantly different ($p = 0.01$), while for the metacarpal muscle, the differences were not statistically significant ($p = 0.2$). The correlation coefficients of the different variables are shown in Table 2. The analysis determined in STATISTICA 12 shows that the lumbar muscle load level is the most correlated with the variables $F$ (N), $A_c$ (mm$^2$), and $p_{max}$ (MPa). In fact, this means that as the individual variables increase depending on the position of the body during the harvest in the order of squatting, kneeling, or keeping straight legs, the level of strain on the lumbar muscles, or the lumbar region of the spine, also increases. In this case, the correlation coefficient is 0.82 for $F$ (N); 0.75 for $A_c$ (mm$^2$), which translates into a strong relationship. For the abductor pollicis muscles, correlation coefficients indicate a quasi-good relationship, at above 0.5. In the case of the metacarpal muscle, for the first two variables, the correlation can be defined at a strong level and in correlation with $p_{max}$ (MPa) at a quasi-good level.

**Table 2.** Correlation coefficient.

| Muscle Tested/Variable | $F$ (N) | $A_c$ (mm$^2$) | $p_{max}$ (MPa) |
|---|---|---|---|
| Abductor pollicis | 0.54 | 0.54 | 0.51 |
| Lumbar right, left | 0.82 | 0.75 | 0.42 |
| Metacarpal muscle | 0.77 | 0.73 | 0.49 |

*3.2. Field Test Results—Surface Pressures*

Figure 10a,b shows the effect of a fixed fruit picking position on the values of the recorded parameters of the Tekscan® system. The highest loads exerted by the picker's index finger were observed during picking in the third—upright—position at 7.87 N, while the lowest were 5.72 N in the first—squatting—position. This was reflected in the maximum surface pressures generated, which increased with the change of working position from 0.168 MPa in position 1, through 0.172 MPa in position 2, to 0.19 MPa for position 3 (upright) (Figure 10b). Variations in the above parameters clearly indicate difficulties in the way strawberry fruit is picked. In the case of the upright position, they are caused by the least comfortable position of the wrist (wrist rotation angle) and reduction in maintaining the stability of the hand position, as well as the depth of inclination of the body posture in comparison with other working positions. The working position was shown to have a significant effect ($p < 0.05$) on the increase in forces and surface pressures exerted by the picker's index finger.

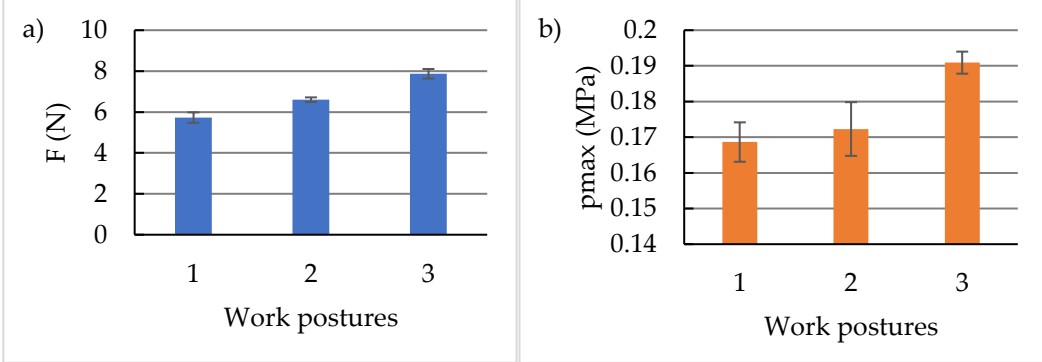

**Figure 10.** Changes in recorded loads (**a**) and maximum surface pressures (**b**) as a function of harvest position (1—squatting position, 2—kneeling position, 3—upright position). The error bars indicate (mean ± SE).

Visualized form of the surface pressures is shown in the contour images in Figure 11. The analysis of the images for the three working positions showed that the highest point pressure concentrations occurred at the contact between the sensor and the achenes evenly distributed on the fruit surface. The recorded images clearly show the change in pressure dispersion on the sensor working surface depending on the position of the finger in a given working position. In the first—squatting—position and second—kneeling—position, due to the easier access of the hand to the fruit and lower inclination of the picker's posture, unevenly distributed areas of surface pressure contours were obtained. In the third—upright—position, on the other hand, the increased discomfort of the work showed an increased but more even accumulation of pressure (increased picking force).

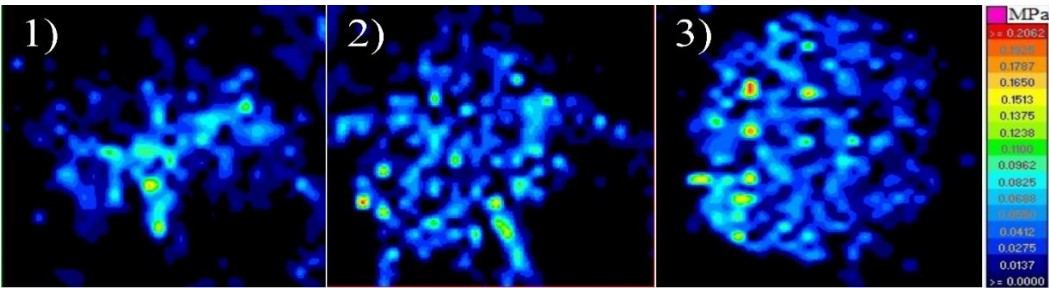

**Figure 11.** Images of surface pressure contours for three picking positions: (**1**)—squatting, (**2**)—kneeling, (**3**)—upright.

Changes in contact between the index finger and the fruit being picked are presented in Figure 12a–c. In the squatting position, both the contact area and the duration of the picking pulse were the smallest (124.1 mm$^2$ and 0.81 s, Figure 12a,b). It is reasonable to believe that the squatting position, requiring less body lean and resulting in a wrist hand position at a lower angle, resulted in increased comfort, positional stability, and speed of picking. A slightly larger surface area and longer contact time between the index finger and the fruit to be picked was required in the kneeling position (151.2 mm$^2$ and 0.87 s), which was undoubtedly related to the different position of the center of gravity and the angular position of the different musculoskeletal members, affecting the speed and repetition of the activity. The variation of the contact area pulse distributions over time for the three harvest positions is shown in Figure 12c. For position 3 (upright), the pulse was characterized by a longer-lasting but rapid increase in contact area values from the moment of contact with the index finger (lasting on average 1.04 s). The analyses carried out showed that the harvesting position had a significant effect ($p < 0.05$) on the increase in the contact area of the index finger with the fruit and the duration of the picking pulse.

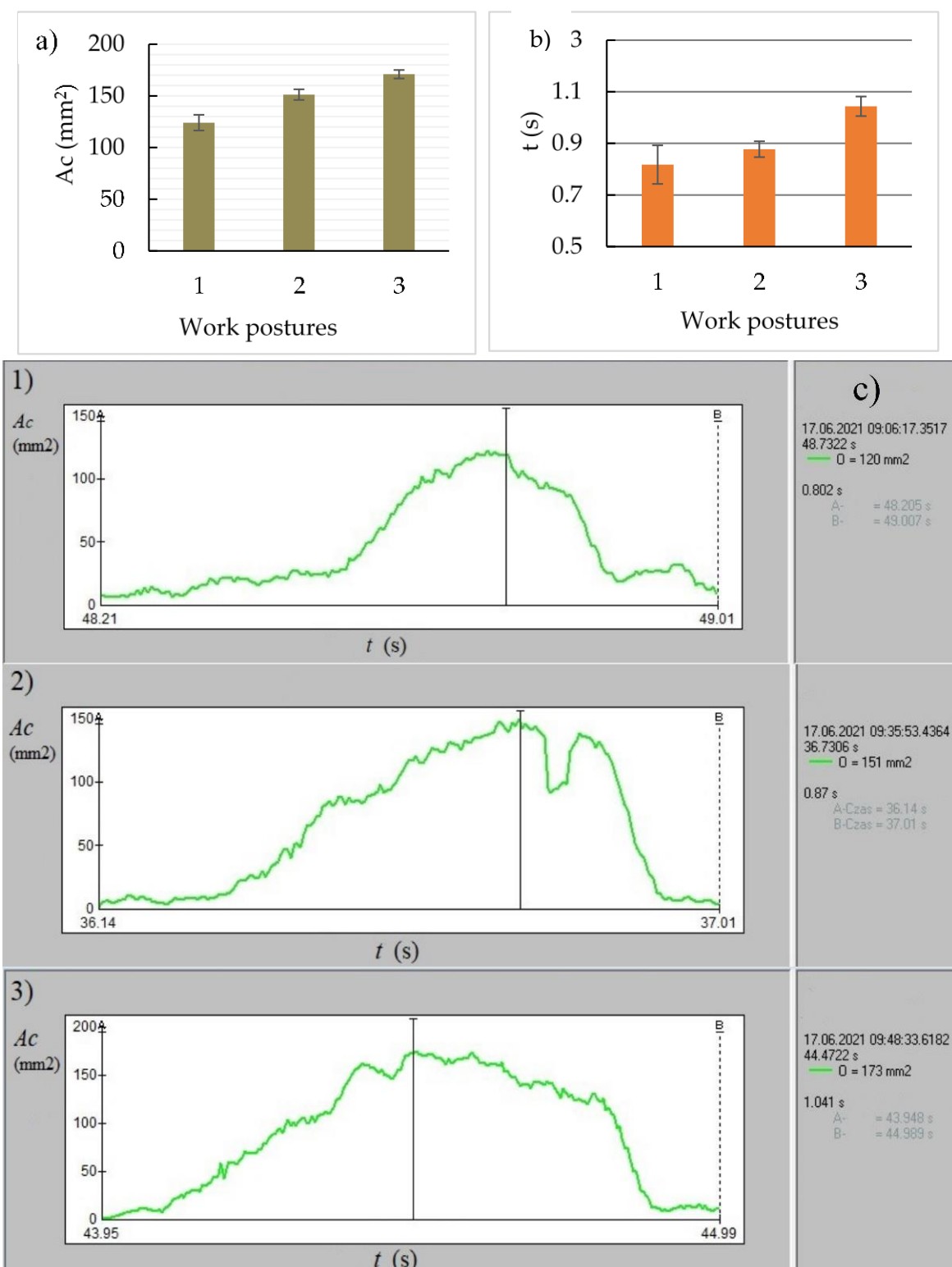

**Figure 12.** Variation of recorded contact parameters for three harvest positions (**1**)—squatting, (**2**)—kneeling, (**3**)—upright: (**a**)—contact area, (**b**)—picking pulse duration, (**c**)—example of contact area pulse waveforms as a function of time $A_c = f(\Delta t)$. The error bars indicate mean $\pm$ SE.

### 3.3. Laboratory Results

Collected material from the field was segregated into four groups, on which laboratory tests were carried out. Three groups consisted of 15 fruits each (geometric and mass measurements), while the fourth group consisted of 45 fruits on which firmness measurements were made. The properties of the test material of the Clery variety listed in Table 3 showed the homogeneity of the selected fruit. The resulting sphericity coefficient deviated from the sphere with a similarity of about 8%. The measured firmness indicated the consumptive maturity of the strawberry fruit.

**Table 3.** Characteristics of analyzed material.

| Cultivar | No. of Fruits | Weight (g) | Mean Diameter (mm) | Height (mm) | Sphericity (-) | Firmness (N) |
|---|---|---|---|---|---|---|
| 'Clery' | 45 | $13.73 \pm 0.25$ | $32.25 \pm 0.52$ | $36.57 \pm 0.46$ | $0.92 \pm 0.01$ | $2.12 \pm 0.18$ |

Each value represents the mean $\pm$ SE of 45 fruits.

The waveforms of load changes as a function of displacement for the whole fruit at three different head speeds of the testing machine are shown in Figure 13. The correlations obtained were characterized by a gradual increase in force up to the maximum values, followed by a gentle decrease. The highest failure loads of 9.7 N were obtained at a strain rate of 100 mm·min$^{-1}$, while the lowest values of 4.1 N were obtained at a quasi-static velocity of 1 mm min$^{-1}$. The above-mentioned differences in load values are due to several factors: with the rapid movement of the loading plate, the fruit flesh cells burst and do not have time to fill the free intercellular spaces, as a result of which the recorded forces reach higher values with a correspondingly large displacement, while the slow loading of the fruit and an incomplete fruit core in the middle allow cell movement and the reduction of destructive forces. It follows from the above that strain rates below 100 mm·min$^{-1}$ are not appropriate for the assessment of the immediate strength of strawberry fruit.

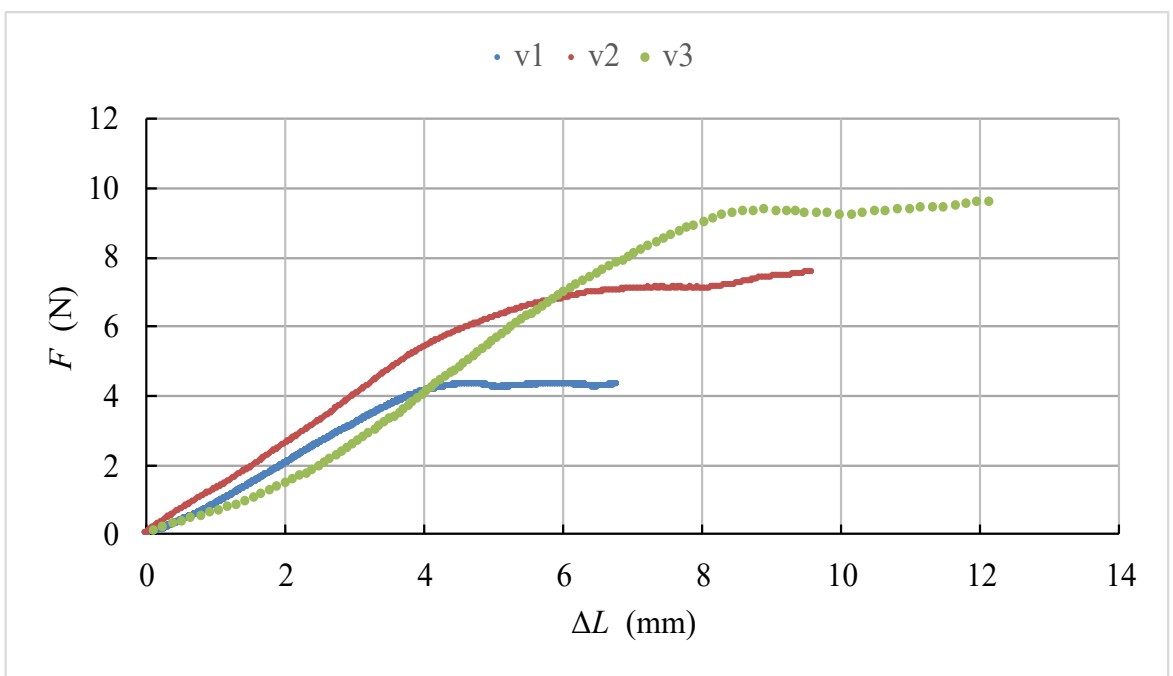

**Figure 13.** Example results of destructive load tests as a function of displacement $F = f(\Delta L)$ for 3 strain rates $v1 = 1$ mm·min$^{-1}$, $v2 = 10$ mm·min$^{-1}$, $v3 = 100$ mm·min$^{-1}$.

Similar changes were observed when interpreting the results of maximum surface pressures in relation to the strain rate (Figure 14). As the strain rate increases from 1 to 100

mm·min$^{-1}$, the maximum surface pressures also increase (from 0.170 to 0.193 MPa) due to the increased load contribution to the compression process.

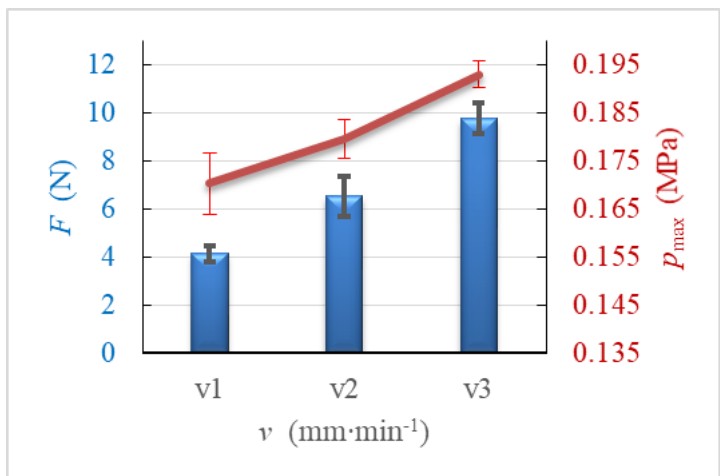

**Figure 14.** Effect of strain rate ($v1 = 1$ mm·min$^{-1}$, $v2 = 10$ mm·min$^{-1}$, $v3 = 100$ mm·min$^{-1}$) on the value of maximum surface pressures and failure loads. The error bars indicate mean $\pm$ SE.

Whole fruit compression tests showed a significant effect ($p < 0.05$) of strain rate on changes in failure load values as well as maximum surface pressures.

Figure 15 shows in a visualized form the images of the changes in the surface pressure contours for the three tested strain rates $v$ (1, 10 and 100 mm·min$^{-1}$). Comparing the above contour images to the images obtained during the three picking positions, it is noted that both the fruit shape and the max surface pressure values for the compression tests are clearer due to the better contact with the non-deformable substrate surface. It is observed that for lower velocities $v1$ and $v2$, the maximum surface pressures are concentrated at the periphery of the contact surface. It is reasonable to believe that this situation was the result of the slow filling of free cell spaces and the presence of an empty fruit core, which resulted in the relief of this area. Image analysis also showed that the quantitative contribution of maximum surface pressures, which are due to the presence of seeds of achenes, increases with increasing strain rate. The achenes are hard and dangerous building blocks of strawberries, which break into the flesh when subjected to high force, causing permanent local damage to the outer tissue structure of the fruit.

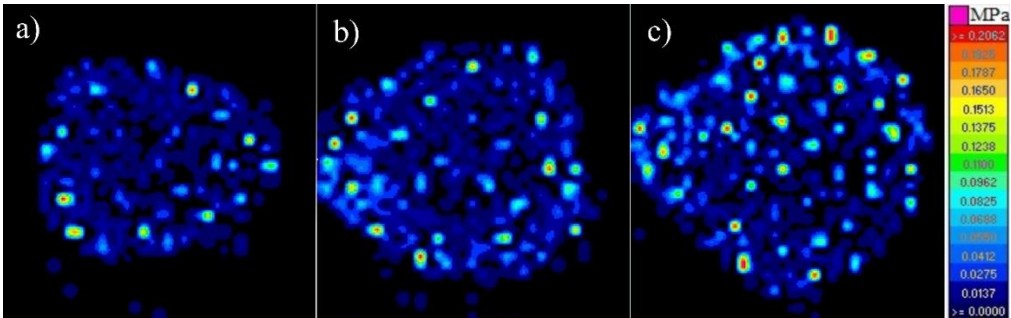

**Figure 15.** Surface pressure contours obtained for three strain rates: (**a**)—1 mm·min$^{-1}$, (**b**)—10 mm·min$^{-1}$, (**c**)—100 mm·min$^{-1}$.

### 3.4. Verification and Comparison of Field and Laboratory Tests

A mutual comparison of the strength parameters recorded during the field and laboratory tests is shown in Table 4. The compression tests carried out were intended to verify how dangerous, from the point of view of material quality, the loads generated during fruit picking in different positions were.

**Table 4.** Comparison of critical strength parameters of field and laboratory tests.

| Type of Test | $F$ (N) | $p_{max}$ (MPa) |
|---|---|---|
| Field (upright pos.) | $7.87 \pm 0.23$ | $0.190 \pm 0.023$ |
| Laboratory ($v3$) | $9.75 \pm 0.63$ | $0.193 \pm 0.028$ |

Each value represents the mean $\pm$ SE.

The field measurements in the third position, i.e., upright, and the destructive tests at the strain rate v3 (100 mm·min$^{-1}$) had the most similar strength parameters. The tests showed that harvesting in an upright position generated loads on the strawberry fruit that were 25% lower than the critical destructive forces recorded in the laboratory, while at the same time having high and comparable values for the maximum surface pressures occurring at the seeds of the achenes (1.5% difference).

## 4. Discussion

The use of sEMG to assess human loads during fruit and vegetable harvesting is currently not common. Usually, ready-made templates, algorithms, or tables are used to assess static or dynamic muscle loading. These methods are characterized by a highly generalized approach to the subject of loads, primarily marginalizing local loads on those segments of the musculoskeletal system that are actually overloaded. High unit overloads in the musculoskeletal system of the worker are neglected, which may translate into under or overestimation of occupational risks. Roman-Liu describes the validity of using surface electromyography (sEMG) in ergonomic research [28]. Kuta reasoned that the level of stress on the picker's musculoskeletal system depends on a number of factors, the most important being the position of the body during work and the location of the fruit on the apple tree [11]. The load level results obtained by using surface electromyography were similar to those obtained by Jakob and Liebers [29] who assessed the load on the musculoskeletal system of workers harvesting and processing fish. Similar values for lumbar muscle loading were obtained in the study by Szubert [30]. The article by Sauter describes an attempt to evaluate the load on the worker's wrists depending on the angle of their deviation from the neutral position, i.e., longitudinal position, in relation to the forearms [31]. As the author notes, the activities performed by the worker should not translate into the need for excessive twisting of the hands, particularly the wrists. According to Fabunmi, the incorrect positioning of the worker's body during lifting and repetitive work is the most common cause of ailments in the human musculoskeletal system and the overloading of this system, which is confirmed by the results obtained in this article [31,32]. The research papers by Barrero and Xiang described a positive correlation between musculoskeletal workload and the uncomfortable body position taken [33,34]. In this article, the authors showed that the deeper the body posture inclination (straight-legged position), the higher the level of muscle strain. Studies have found that men are more likely to suffer from these conditions than women. Men have a wider range of jobs and are therefore more exposed to musculoskeletal disorders. Buchle stresses the need to continue research into improving worker safety, particularly as regards exposure to musculoskeletal disorders caused by dynamic loading [35].

The results of these studies indicate that the strawberry is a very sensitive fruit and is not resistant to sudden compressive loads. Mechanical damage to the fruit is a defect in the biomaterial and is closely related to the mechanics of the fruit [36]. Nagata showed that subjecting the fruit to a compressive force of more than 2 N can initiate bruising on the surface of the strawberry after only 48 h [37]. Aliasgarian reports that the shape of the strawberry fruit is a factor leading to the susceptibility of the fruit to damage, due to its spherical values [2]. The authors reported that conical and oblong strawberries subjected to compression pressure were firmer and more resistant than large and more spherical strawberries. The characteristics of the Clery variety showed that the collected material did not have greatly elongated shapes (sphericity of 0.92), and thus it can be thought that this additionally influenced the lower resistance of the strawberries under compressive loads both during picking and destructive tests. In their study, the authors used a modern method

for evaluating the strength properties of strawberries based on the direct measurement of index finger pressure on the fruit surface in the range of loads occurring during picking. At present, there is a lack of similar experiments by other researchers to compare the results obtained for strawberries. For apples, Młotek showed that, during the rotation technique of picking, the index finger, in addition to the thumb, makes the greatest contribution to the load transfer [10]. Their compression tests on five apple varieties also showed that, within the range of permissible surface pressure values, the danger of causing permanent mechanical damage to the fruit could occur in the average pressure range of 0.1 to 0.2 MPa. Furthermore, Kuta demonstrated that changing the height of apple picking affects the level of picker muscle tension and the surface pressure exerted by the hand on the fruit [11]. The highest values of average pressures generated by the thumb were observed at extreme picking heights of 0.5 m (0.09 MPa) and 2 m (0.11 MPa).

In the literature, measurements of forces and surface pressures in compression, quasistatic loading, dynamic loading, or impact tests have been carried out for both single and layer-transported strawberry fruit. Ferreira exposed the fruit to continuous compression at different temperatures using the IFAS Firmness Tester [38]. They showed that fruit hydrocooled at a low temperature (1 °C) was more resistant to compression, while fruit at a higher temperature (20 or 24 °C) was more resistant to impact. Similar to the authors of this work, firmness at 2.5 N was also obtained by Wei for the strawberry variety HongFyan [1]. The evaluation of the mechanical properties of strawberries under the influence of different doses of pulsed light at 6 °C was carried out by Duarte-Molina [39]. In their study, they demonstrated cell wall strengthening and significant subcutaneous cell wall integrity induced by pulsed light stress. In fruit puncture tests at 30 mm·min$^{-1}$, they obtained similar force values in the range of 5–7 N to the authors of this paper. Compression tests on Hongyan strawberries were conducted by An, who, at strain rates ranging from 60 to 300 mm·min$^{-1}$, obtained peak loads as a function of percent strain (in terms of local plastic strain) of up to 20 N in their study [40]. Their research also showed a similar increase in load with increasing strain rate as confirmed in Figure 8. The authors of this paper obtained a similar characteristic of the curves, while the differences in the measured loads compared to the results of An were mainly due to the different method of measurement (compression to a specific strain along the radius) and the fact that the fruit was harvested at bright red maturity. The authors also observed that the soft pad of the index finger did not generate as high a load as when the fruit came into contact with the rigid bed of the stationary machine component when picking the strawberry fruit. It should be noted that, due to the force concentrated on a small contact area, a more flexible substrate absorbs impact energy and can also reduce mechanical damage, as confirmed by Li and Thomas and Li [35,41]. In their results on bruises of Golden Delicious apples, Komarnicki showed a more than twofold reduction in damage generated on the soft polyethylene foam bed [42]. This work is a combination of an ergonomic approach and fruit harvesting techniques as well as fruit biological properties. The work methodology and conception can be used for research while harvesting other fruits or vegetables as well as in many areas of industry or agriculture.

## 5. Conclusions

The simultaneous measurement of surface pressure and the corresponding recording of the sEMG used in this study provides an innovative way of assessing both the risk of damage to the fruit being picked and the risk of overloading the body in a given working position, which may prevent musculoskeletal injuries to the picker. The authors proved that working position has a significant effect on the increase in forces and surface pressures exerted on the fruit by the picker's index finger. Picking in the squatting position proved to be the most comfortable for the picker and the safest for the quality of the harvested material. Picking in an upright position, on the other hand, increased the forces, pressures, contact surface, and time required to pick the strawberry fruit, which was due to the uncomfortable positioning of the wrist and a reduction in maintaining the stability of

the hand position, as well as the depth of inclination of the body posture compared to other working positions. Whole fruit compression tests showed a significant effect of strain rate on changes in failure load values as well as maximum surface pressures. As the deformation velocity increases, the maximum surface pressures increase, which is related to the overly slow filling of the intercellular voids and thus the rapid load contribution to the compression process. The analysis of the contour images also showed that the quantitative contribution of maximum surface pressures increases with increasing strain rate, which is due to the presence of hard achene seeds. The compression tests carried out allowed the authors to verify how dangerous, from the point of view of material quality, the loads generated during fruit picking in different positions were. The tests demonstrated that the least comfortable upright harvesting position generated loads 25% lower than the critical destructive forces recorded in the laboratory, but with comparable values for the maximum surface pressures occurring around the achene seeds. Consequently, harvesting in an upright position is not recommended to maintain high fruit quality and the efficiency and fitness of the picker.

The sEMG tests confirmed that the highest level of stress during strawberry picking occurs when the picker is standing on straight legs. In this position, the highest loads are generated by the lumbar muscles, where the % MVC value exceeds 30%. The lowest loads occur for the metacarpal muscle as they do not exceed 5% of the MVC in any of the three positions analyzed. The surface electromyography technique used proved to be a very accurate and precise method of load measurement compared to traditional tabular methods. Therefore, the applied research concept offers the possibility of improving working conditions, taking ergonomic considerations into account; on the other hand, this can be combined with the Tekscan® system. This enables the definition of model conditions for obtaining biological material without disturbing its structure during harvesting. From the perspective of further research, it is possible to use a more extensive system for ergonomic analysis, including a module for spatial work analysis with defined boundary conditions, among others.

**Author Contributions:** Conceptualization, P.K. and Ł.K.; methodology, P.K. and Ł.K.; software, P.K. and Ł.K.; validation, P.K. and Ł.K.; formal analysis, P.K.; investigation, P.K. and Ł.K.; resources, P.K.; data curation, Ł.K.; writing—original draft preparation, P.K.; writing—review and editing, Ł.K.; visualization, P.K.; supervision, P.K.; project administration, Ł.K. All authors have read and agreed to the published version of the manuscript.

**Funding:** This research received no external funding.

**Institutional Review Board Statement:** Ethical review and approval were waived for this study, because the research was performed only by authors who gave their written approval for this research.

**Informed Consent Statement:** Written informed consent has been obtained from the patient(s) to publish this paper.

**Data Availability Statement:** The source data presented in this study are available on request from the corresponding author. The data are not publicly available due to their further processing and use in model building.

**Acknowledgments:** The research is co-financed under the Leading Research Groups support project from the subsidy increased for the period 2020–2025 in the amount of 2% of the subsidy referred to Art. 387 (3) of the Law of 20 July 2018 on Higher Education and Science, obtained in 2019.

**Conflicts of Interest:** The authors declare that they have no known competing financial interests or personal relationships that could have appeared to influence the work reported in this paper.

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
