# Peer review of "Evaluation of Picker Discomfort and Its Impact on Maintaining Strawberry Picking Quality"

_applsci, doi:10.3390/app112411836_

Round 1

Reviewer 1 Report

- Fig. 12 b is missing
- Some English language errors must be corrected (ex. The the (line 31), to study ->studying (line 66), the of -> of the (line 70), use quasi-static consistently all over the document)
- It is not clear from the Material and Method section or from Results if the load of the test muscles was recorded only for 60 s or a window of 60 seconds was selected for exemplification of load waveforms for different picking positions
- have the authors taken into consideration the influence of physical dimensions of the worker (height, if it is slim or fat, if they have thin or thick fingers etc) on the mucle loads and thus on the picking force applied to the fruit? How passing time influence these parameters?
- most of the comparison on measuring methods and effects of force applied to the fruit is made with apples. Apples and strawberries have very different size, structure, damage evolution, stress resistance etc. Are there similar studies carried on fruit with characteristics closer to strawberries?

Author Response

Answer to Reviewer’s comments

Fig. 12 b is missing

Answer: The graph was skipped when converting file to pdf. The problem has now been resolved.

- Some English language errors must be corrected (ex. The the (line 31), to study ->studying (line 66), the of -> of the (line 70), use quasi-static consistently all over the document)

Answer: These errors were corrected.

- It is not clear from the Material and Method section or from Results if the load of the test muscles was recorded only for 60 s or a window of 60 seconds was selected for exemplification of load waveforms for different picking positions

Answer: These infromations (graphs) were corrected.

- have the authors taken into consideration the influence of physical dimensions of the worker (height, if it is slim or fat, if they have thin or thick fingers etc) on the mucle loads and thus on the picking force applied to the fruit? How passing time influence these parameters?

Answer: Yes, during our studies, the anthropometric data were determined.

- most of the comparison on measuring methods and effects of force applied to the fruit is made with apples. Apples and strawberries have very different size, structure, damage evolution, stress resistance etc. Are there similar studies carried on fruit with characteristics closer to strawberries?

Answer: We agree with the Reviewer, research on apples is published so often, which may be due to the high availability of these fruits and a fairly homogeneous structure of the material. Strawberry, on the other hand, is a difficult research material – as a soft fruit, which is characterized by high susceptibility to damage, short harvesting period, short durability, rapid loss of quality after harvesting, but there are some reports that describe their properties:

https://doi.org/10.1016/j.jfoodeng.2020.110016

https://doi.org/10.1515/ata-2015-0001

https://doi.org/10.1111/jfpe.12207

On the other hand, there is a lack of research in the literature linking the issues of harvesting ergonomics with the quality of the obtained soft fruit, including strawberries.

Reviewer 2 Report

The article deals with an interesting way of evaluating strawberry picker comfort and how the working position influences the quality of the harvested fruit. The surface EMG acquired from three locations was used for the assessment of musculoskeletal system load. In addition, the surface pressure measurement system by Tekscan was used to determine the surface pressure exerted on the strawberry during picking. The combination of these systems and evaluation methods are innovative and original. The conclusions are consistent with the results provided in the graphs and tables. The references are in line with the topic of the article.

Although the paper is clearly written, I have the following comments on the article in order to enhance its quality:

  • Figure 1 shows the measurement system placed on the hand. It is advisable to describe parts of the system in the picture. Could you provide the placement of surface pressure sensors in a detailed image? It would be more demonstrative and evident.
  • The used equipment source is not in a uniform and correct format, e.g. Line 164: State name of EMG Software (manufacturer, city, country); Line 204; Line 206; Line 217; etc.
  • Line 255: “ The results are presented as a percentage of maximal muscle strength (% MVC)“. Could you specify how the maximal muscle strength (100 %) was determined?
  • The graphs in Figures 7, 8, 9, 13 will look more professional if the data lines are thinner. Consider using a vertical grid for better readability.
  • Could you denote the moments of each strawberry detaching from the stem in Figures 7, 8 and 9?
  • Please add a horizontal grid to the graphs in Figure 10.
  • Figure 12 is corrupted and chaotic. The graph in b) section is not shown (problem with pdf conversion?). The c) section is not denoted as c); it includes areas 1), 2), and 3), which are not described in the figure description. The c) section should have the same style as other graphs.
  • Please use mm.min-1 instead of mm (space) min-1

Author Response

Answers to the Reviewers

The article deals with an interesting way of evaluating strawberry picker comfort and how the working position influences the quality of the harvested fruit. The surface EMG acquired from three locations was used for the assessment of musculoskeletal system load. In addition, the surface pressure measurement system by Tekscan was used to determine the surface pressure exerted on the strawberry during picking. The combination of these systems and evaluation methods are innovative and original. The conclusions are consistent with the results provided in the graphs and tables. The references are in line with the topic of the article.

Although the paper is clearly written, I have the following comments on the article in order to enhance its quality:

  • Figure 1 shows the measurement system placed on the hand. It is advisable to describe parts of the system in the picture. Could you provide the placement of surface pressure sensors in a detailed image? It would be more demonstrative and evident.

Answer: Thank Reviewer for opinion, we have corrected the graph number 1, which shows the contact of the index finger with the pressure sensor.

  • The used equipment source is not in a uniform and correct format, e.g. Line 164: State name of EMG Software (manufacturer, city, country); Line 204; Line 206; Line 217; etc.

Answer: We have added these information.

  • Line 255: “ The results are presented as a percentage of maximal muscle strength (% MVC)“. Could you specify how the maximal muscle strength (100 %) was determined?

Answer: At the beginning of the research, the maximum value of muscle tension was determined (100% MVC of the muscle). The pickers flexed the muscle to its maximum value at this point.

  • The graphs in Figures 7, 8, 9, 13 will look more professional if the data lines are thinner. Consider using a vertical grid for better readability.

Answer: To improve readability, the vertical grid lines in Figures 7, 8, 9, 13 have been added and the thickness of the data lines has been reduced.

  • Could you denote the moments of each strawberry detaching from the stem in Figures 7, 8 and 9?

Answer: We added the moments in which the strawberries were picked on the graphs.

  • Please add a horizontal grid to the graphs in Figure 10.

Answer: The horizontal grid lines in Figure 10 have been added to improve its readability.

  • Figure 12 is corrupted and chaotic. The graph in b) section is not shown (problem with pdf conversion?). The c) section is not denoted as c); it includes areas 1), 2), and 3), which are not described in the figure description. The c) section should have the same style as other graphs.

Answer: Thank you for your comment, we agree that there was a problem with converting file to * pdf format, the figures 12a and 12b have been added and corrected, section c) has been added, in which the authors, however, cannot edit it because the graphs are generated automatically by the program and it is not possible to interfere with the style of the chart. The charts in section c) are original.

  • Please use mm.min-1 instead of mm (space) min-1

Answer: these records have been corrected.

Reviewer 3 Report

Overview
======================
In this paper the Authors present a relationship between the assumptions of ergonomics postures at work of a strawberry picker and the quality of picked fruit.  From the tests it follows that the more comfortable position of the worker's body the less negative is the effect on the quality of the harvested fruit. 

General comments
======================
- The work is undoubtedly really original and I think that handling a data process from the strawberry fields to the lab was not easy to handle with.
However, with respect to ergonomics problems related to the worker the problem of preserving the strawberry seems to me to be secondary.  I really like the idea to analysed data such to correlate Bothe the aspects (ergonomics-fruit preservation) but I think that the entire paper has to be human-centered and not fruit-centered.  This would make the paper more versatile and usable for many purpose evaluations.
- Concerning the use of the EMGs, looking at the Fig1 it seems that they are quite invasive.  Do you think that this may lead to a discomfort and a different picking execution?  The preservation of the fruit could be affected from this?

Minor comments
======================
- Check the (many) grammar typos in the text.

Author Response

Reviewer comments

- The work is undoubtedly really original and I think that handling a data process from the strawberry fields to the lab was not easy to handle with.
However, with respect to ergonomics problems related to the worker the problem of preserving the strawberry seems to me to be secondary.  I really like the idea to analysed data such to correlate Bothe the aspects (ergonomics-fruit preservation) but I think that the entire paper has to be human-centered and not fruit-centered.  This would make the paper more versatile and usable for many purpose evaluations.
- Concerning the use of the EMGs, looking at the Fig1 it seems that they are quite invasive.  Do you think that this may lead to a discomfort and a different picking execution?  The preservation of the fruit could be affected from this?

Answer

Thank you for your favorable opinions. It is a pilot paper in this field of research, therefore we initially assess whether such study is interesting. Of course, we will, focus on human ergonomics area in our next studies.

In this study a non-invasive EMG method was used. The electrodes were put on the skin above the muscles in order to determine a muscle load.

In medicine, for example there is an invasive method used,  The examined muscle is tested with needle. The weight of the preamplifiers is about 2 grams. It is not a physical load.

Round 2

Reviewer 3 Report

I really appreciate the modifications the Authors did on the manuscript.

Author Response

Dear Reviewer,

Thank you very much for your comment.

We have checked and corrected the manuscript in aspect of language.
